# The Influence of Deformation under Tension on Some Mechanical and Tribological Properties of High-Density Polyethylene

**DOI:** 10.3390/polym11091429

**Published:** 2019-08-31

**Authors:** Maciej Kujawa, Piotr Kowalewski, Wojciech Wieleba

**Affiliations:** Wroclaw University of Science and Technology, Faculty of Mechanical Engineering, Wybrzeze Wyspianskiego 27, 50-370 Wroclaw, Poland

**Keywords:** friction, wear, hardness, surface free energy, stress, strain

## Abstract

Polymer materials are increasingly being used for sliding machine elements due to their numerous advantages. They are used even where they are deformed and in such a state that they interact frictionally, e.g., in machine hydraulics or lip seals. Few publications deal with the influence of deformation, which is the effect of, e.g., assembly on tribological properties of polymeric material. This deformation can reach up to ε *≈* 20% and is achieved without increasing the temperature of the polymer material. The paper presents the results of investigations in which high-density polyethylene (PE-HD) was maintained in deformation by means of a special grip (holder). The wear of the sample was significantly higher than that of the undeformed sample. This effect persisted even after partial relaxation of the stress in the sample after 24 h. Additional investigations were carried out to explain the obtained results. There were the microscopic observations of the surface after friction, measurements of microhardness, and surface free energy. Changes in the value of surface free energy and a significant decrease in microhardness with deformation under tension were observed. Deformed materials have a different surface appearance after friction and a different size and form of wear products. It was indicated that it is probable that the cohesion of the material will decrease and that the character of the wear process will change as a result of tension. Deformation under tension without heating of polymeric material (PE-HD), e.g., as a result of assembly, has been qualified as a threat to be taken into account when designing and analysing polymeric sliding elements.

## 1. Introduction

Polymer-based sliding elements are increasingly being used in mechanical engineering. This is a result of their favourable properties such as a possibility to work without lubrication, low coefficient of friction when working with steel, high chemical resistance, easy shaping, and low cost of producing elements, especially in mass production. Polymer sliding elements also appear where they are deformed during tribological interaction. Stryczek et al. conducted tests on the use of polymer materials in machine hydraulics [1]. The result of this work is, among other things, a prototype of a hydraulic cylinder. As a result of the oil pressure, the cylinder is deformed and, in this state, it interacts frictionally with the seal (Figure 1a). Another example is PE-UHMW (ultra-high molecular weight polyethylene) wires used in spine stabilisers [2]. The wiring subjected to stress and strain maintains the vertebrae in position and interacts frictionally with the metal rods during the patient′s movements (Figure 1b). Another example can be lip seals, which after mounting on the shaft are deformed and, in this state, interact frictionally with the metal surface (Figure 1c). Sliding elements made of polymeric materials may also be used in modern underwater robots where they are deformed as a result of hydrostatic pressure [3].

In the literature, there are many publications describing attempts to obtain a polymeric material with more favourable tribological properties by tension. This type of treatment consisted in the introduction of significant deformation values amounting to several hundred percent. They were carried out by drawing polymeric plates at elevated temperature [4,5,6] during extrusion or at room temperature by stretching the samples on a tensile testing machine [7,8,9]. The abovementioned studies showed that the polymeric material subjected to tension has a different coefficient of friction and wear coefficient in interaction with steel compared to the undeformed material.

Few works concern the relatively low value of the deformation obtained at room temperature, i.e., the situation corresponding to the deformation resulting from the assembly. Cayer-Barrioz et al. studied the interaction of a steel roll with a strained polymeric rope [10]. The polymer lip seals, which are strained as a result of shaft mounting, have received a great deal of attention as they are often used in mechanical engineering. A lot of studies and simulations have been developed and presented, in which strain and stress in shaft-seated seals have been determined [11,12,13,14]. Their aim was to obtain the possibility of forecasting frictional resistance and durability of seals. In these analyses, however, the possible influence of deformation on tribological properties of the polymeric material was not taken into account.

The authors decided to investigate the influence of cold deformation on the tribological properties of polymeric materials for four main reasons: firstly, increasing the number of polymer-sliding elements appearing where they are deformed during tribological interaction. Secondly, there is few information in the literature about the influence of cold deformation on the tribological properties of polymeric materials. Thirdly, the phenomena and processes occurring during the cold deformation of polymeric materials can influence the friction and wear of the polymeric material [10]. Finally, an additional premise behind the choice of this problem is the fact that the influence of deformation on tribological properties was indicated for other materials—metals [15]. Therefore, the authors decided to investigate the influence of cold deformation on the tribological properties of polymeric materials. The authors already signalled this issue in earlier publications [16,17] and have now prepared a set of studies focusing on PE-HD—a polymeric material that proved to be sensitive to tension.

## 2. Materials and Methods

The works carried out included tests of wear, microhardness, and surface free energy. The results of tribological tests showed that the wear of PE-HD significantly increases after the polymer material is deformed by tension. The studies on microhardness and surface free energy were designed to provide information that would help to explain the behaviour of the polymer. In order to obtain information on the influence of deformation on the adhesive component, tests of surface free energy were carried out. Microhardness measurements helped to evaluate the influence of deformation on the cohesion of the polymeric material.

### 2.1. Samples

In the studies, high-density polyethylene PE-HD (PE 300) Polystone^®^ G AST natural (Röchling, Haren, Germany) was used. Preliminary investigations have shown that this polymeric material is susceptible to deformation under tension and that its wear after applying tension significantly increases. This polymer is used for the production of sliding elements such as bushings and seals.

The samples for tribological tests during tensile testing had a shape similar to the outline of the so-called “paddle” presented in the EN ISO 527:2012 standard for the determination of tensile properties of plastics. As the space at the test stands was limited, the shape of the sample was reduced and slightly modified. The research was carried out during frictional interaction in the direction parallel and perpendicular to the direction of the tensile force vector. Therefore, two types of samples suitable for both situations were prepared (Figure 2). Samples were cut from 2-mm thick plates. The gripping sections of the samples placed in properly profiled callipers ensured that the samples were securely clamped. Measurements were made in the measuring section distant from the grip section to avoid analysis of the material in the notch impact area.

### 2.2. Maintenance of Deformation

When a polymer sample is strained at room temperature, e.g., on a tensile testing machine, and then removed from the grip and left freely, the deformation may change. This is due to the viscoelastic properties of the polymeric materials. Depending on the type of polymer and the strain value, the sample can be fully or partially restored to its original form. Sliding elements and seals remain deformed after assembly. To reproduce this situation, the polymer samples ought to remain in a deformed state during the test; therefore, they had to be held in grips to prevent them from returning to their original form.

Special handles with PLA (polylactide) callipers were designed and manufactured in three-dimensional printing technology (Figure 3). The callipers had sockets to ensure that the sample is locked in place. The shape of the grip (holder) was adapted to each test stand in order to avoid collisions with the elements of the measuring instrument.

The changes in callipers spacing and, thus, in the tension of the sample were achieved by turning the nuts on the bolt. The nuts in the tension grip were turned manually. Therefore, it is difficult to precisely determine the deformation rate of the samples. Based on the time during which the deformation took place (turning the nuts) and the obtained value of the deformation, it can be estimated that the deformation took place at a speed of about 5–10 mm/min.

### 2.3. Deformation Measurement

The samples used in the study had the shape of so-called “paddles”. The part between the grip and the measuring part was rounded. Due to variable stiffness along the length of the sample in areas adjacent to the gripping parts, the strain after tension was different from that at the centre of the measuring part. For this reason, deformation was not determined by the distance between the callipers but was checked at the place where it was intended to be measured during tribological tests.

The value of deformation was determined by measuring the distance between the imprints of the indenter used to measure microhardness using the Knoop method. The imprints were oriented in a direction perpendicular to the direction of the tensile force vector. The initial distance between markers (Figure 4) was determined for an undeformed sample. Subsequently, the measurement was repeated after the sample deformation. The difference in distance between the markers allowed to determine strain *ε*. This way of determining the strain allowed us to obtain a precise value of the strain in the place where further measurements were made, i.e., microhardness, surface free energy, or wear.

The tests were carried out to determine the deformations that could occur during assembly. Four values of tensile strain were adopted: 2%, 5%, 20%, and 50%. For deformations below ε = 2%, in the polymers, elastic deformations prevail. According to the recommendations, the deformation of plastic parts in machines and equipment should not exceed 2%. Therefore, a deformation range of up to 2% can be defined as acceptable, tolerable, and safe. A 5% strain is outside the acceptable range. Such deformation may occur, among other things, in an improperly mounted element. Tensile strain of 20% can occur during assembly in some machine parts, e.g., in the lip seal of the shaft. In the terms of 50% strain, it is difficult to find a machine element in which such deformation would be the result of the assembly. However, this deformation may be achieved by deliberate deformation of the sliding element. Tests conducted for 50% strain can provide information on whether cold deformation significantly affects the tribological properties of the analysed polymer and whether it can be used as a technology used in the production of sliding elements. In addition, measurements taken for such a high deformation may contribute to a deeper explanation of the friction and wear of deformed machine elements.

### 2.4. Measurement of Wear

Wear tests were carried out on a test stand in the combination of a roll and a block. The rotating steel roll worked with a polymer sample kept in a deformed state (Figure 5). The volume of material removed was calculated based on the roller’s dimensions and the measured length of the groove created in the polymer sample (according to norm ASTM G77-17).

The step motor shaft was equipped with a 30-mm diameter and 3-mm wide stainless steel X8CrNiTiS18-9 roller. The roughness of the cylindrical surface of the roll was Ra 0.85 ÷ 1.25 µm (*C*_t_ = 0.25; *L*_m_ = 1.25). The speed was 0.33 m/s.

The load on the roll was selected so that, at the end of the interaction (when the groove was about 5 mm-long), the average pressure was about 3 MPa. The load on the roll was 49 N. The friction path was selected experimentally so that the length of the groove during the tests does not exceed the assumed value of 5 mm. This length of groove ensured a sufficient distance between the roll and the edge of the sample. The friction path was 2000 m.

As the wear coefficient *K*_w_ is often used by manufacturers of sliding materials and given in the product documentation, it was decided to present the wear results using this ratio. The *K*_w_ indicator is determined by the following formula:(1)$$K_{w}=\frac{V}{F_{N}S}$$
where *V* is the volume of worn material (mm^3^), *F*_N_ is the normal force (load) during friction (N), and *S* is the friction path (m).

### 2.5. Microhardness

The results of the wear tests show that the direction in which frictional interaction was carried out was important for the deformed polymer. Therefore, a method that would provide information depending on the setting of the indenter was sought for hardness testing. The position of an indenter relative to the material is important in the Knoop method because the indenter is a diamond-shaped pyramid (Figure 6). When measuring the imprint, the size of only one (longer) diagonal is determined.

Shimadzu HMV-2 microhardness tester was used during the research. The measurements were carried out by maintaining the samples in a deformed state using a grip (Figure 7). For the Knoop microhardness measurements, the lowest load (98.07 mN) and the lowest load time (5 s) were used. The measured diagonal lengths of the indentations ranged from about 170 µm (for undeformed material) to about 220 µm (for deformed material). Using the known angle of the indenter (172°30′) and trigonometric relationships, the depth of the imprint after the removal of the indenter was calculated. It ranges from approximately 5 to 7 µm.

The measurements included tests in the parallel and perpendicular directions to the tensile force direction. Due to the fact that the results of microhardness tests turned out to be very significant, the tests for more strain values (0%, 2%, 5%, 20%, and 50%) were conducted.

### 2.6. Surface Free Energy

In order to evaluate adhesion in the combination of deformed polymer–steel, tests of surface free energy were carried out. The measurements were taken directly after applying tension to achieve 20% and 50% strain and after 30 min of maintaining the deformation ε = 50%. The Kruss DSAHT12 goniometer was used in the study. Drops of four liquids were applied on the polymer sample: distilled water, diiodomethane, ethylene glycol, and formamide. The angle between the droplet outline and the surface of the polymer sample was measured using a grip to maintain the sample deformed. The value of surface free energy was determined using the Owens–Wendt–Rabel–Kaelble (OWRK) method.

### 2.7. Observation of the Surface by Means of SEM

Surfaces of interacting elements are subject to different changes resulting from processes and phenomena occurring on the contact surface during friction. The condition of the surface subjected to friction can give a lot of information helpful in explaining them; therefore, it was decided to include microscopic observations in the research. The investigations of sliding surfaces of polymeric samples after interaction with a steel roll were carried out with the use of a scanning electron microscope (SEM).

The space in which the sample was placed was very limited. The grips described above were too large to fit into the microscope socket. The polymer was kept stretched during microscopic observations owing to a special base produced using a 3-D printer (Figure 8a).

The sample was stretched in a standard grip (Figure 8b). Two holes were made in the measuring part; the polymer was placed in the grip and fixed with nuts (Figure 8c). Then, the measured part was cut off from the gripping parts and the whole was placed in the microscope slot (Figure 8d).

A drawback was observed in the method described above. Namely, after the procedure of transferring the sample to a smaller grip, the deformation was reduced by “removing the clearance” between the screw and the inner wall of the hole in the polymer. Measurements showed that, for initial sample deformation ε = 20% after cutting off the gripping parts, the final deformation was about 17–18%. Despite the decrease in strain value, microscopic observations showed differences in surface appearance after the process of friction between deformed and undeformed materials. The assessment of the surface condition after friction was carried out for an undeformed polymer and after stretching to 20% strain.

### 2.8. Statistical Analysis of the Results

In terms of tests for each variant (parallel/perpendicular direction + strain value), the measurements were carried out for four polymeric samples. Immediately after the deformation, the sample together with the grip was mounted on the test stand. If the grip was placed on the stand incorrectly (e.g., in a slight incline), the result of the measurement was significantly different from the others. The Dixon test (Q test) was used to eliminate such results (vitiated with a gross error). Other results were used to calculate the mean value (x) and the expanded uncertainty (*U*):(2)$$U=\frac{k\times S}{\sqrt{n}}$$
where k is the coverage factor (the value of k = 2 used determines the probability of finding the actual value within ±*U* of 95%), *S* is the standard deviation, and *n* is the number of measurements. The results of the research were given in a form x ± *U*.

This paper focuses on the comparison of the behaviour of a deformed polymeric material with that of an undeformed polymer material. Therefore, by presenting the results, the percentage change of wear coefficient, microhardness, and surface free energy were calculated:(3)$$\Delta K_{w}=\frac{K_{w\varepsilon }-K_{w}\;}{K_{w}}$$
where *K*_w_ is the wear coefficient for the undeformed polymer and *K*_wε_ is the wear coefficient for the deformed polymer.
(4)$$\Delta HK0,01=\frac{HK0.01_{\varepsilon }-HK0.01\;}{HK0.01}$$
where *HK0.01* is the microhardness of the undeformed polymer and *HK0.01_ε_* is the microhardness of the deformed polymer.
(5)$$\Delta SFE=\frac{SFE_{\varepsilon }-SFE\;}{SFE}$$
where *SFE* is the surface free energy of undeformed polymer and *SFEε* is the surface free energy of the deformed polymer.

## 3. Results

### 3.1. Impact of Deformation Direction on PE-HD Wear

After analysing the results of wear measurements, it turned out that the expanded uncertainty range in most cases covered the range of ±10% of the average value. Therefore, the change was considered significant when it was greater than or equal to 20%. The tables’ cells highlighted in orange present the results for which the mean value for a given deformation differed by 20% or more from the mean value for an undeformed polymer. The results are divided into two groups obtained during frictional interaction in the directions parallel and perpendicular to the direction of the tensile force vector.

The results showed a significant effect of tensile stress on the wear of the polymer material when the direction of friction was parallel to the direction of the tensile force vector. In the terms of 20% strain, the *K*_w_ wear coefficient was 5.5 times higher than in the terms of undeformed polymer (Table 1 and Figure 9). With such deformation (ε = 20%), the wear coefficient was the highest. Further increase in deformation to 50% strain led to a reduction in the wear coefficient. In addition, a significant increase in the wear coefficient (51%) was already recorded for 2% strain.

When the direction of friction was perpendicular to the direction of the tensile force vector, the deformation also affected the polymer wear (Table 2 and Figure 10). Significant effect of tension on PE-HD wear was observed from 2% strain. After lengthening, the wear coefficient increased the most at ε = 20% (increase by 289%).

The results showed that the deformation of polymeric material has a significant influence on its wear. The wear coefficient (*K*_w_) increases as the deformation increases.

In summary, it can be concluded that PE-HD polyethylene is very sensitive to tension. Differences depending on the direction of friction were visible. When working in the direction parallel to the direction of the tensile force vector, the changes in the wear coefficient were even several times greater than in the perpendicular direction. For 2% strain, the differences between the results for the two directions were still small, whereas for ε ≥ 5%, they were clearly visible.

#### 3.1.1. Impact of 24-Hour Maintenance of Tension on Wear

Seals and sliding elements made of polymers are deformed during assembly. During operation, the deformation practically does not change. At room temperature, the polymers, which are viscoelastic materials, undergo stress relaxation. Significant differences in wear occurred for the test samples immediately after tension. Due to stress relaxation after a certain period of time after deformation, the polymer wear may be different.

Since time is an important parameter in the analysis of deformed polymeric materials, additional tribological investigations were carried out. The aim was to explain how the materials tested for the purposes of this study will behave after a certain period of time. The polymer samples were deformed and left in the grip for 24 h. During the period up to 24 h, stress relaxation occurs intensively and the stress in the polymeric material decreases. The samples were stretched to 20% strain as the observed changes in wear were significant at this value in the previous stage of the research. After 24 h from the introduction of the deformation, wear tests were carried out, in which the friction was in the direction parallel to the direction of the tensile force vector.

The obtained results showed a significant change in the PE-HD wear coefficient. Comparing the results (Table 3 and Figure 11) for ε = 20% deformation immediately after stretching and the results 24 h after stretching, the PE-HD wear coefficient 24 h after stretching is approximately 50% higher than for immediate deformation and 879% higher than for undeformed samples.

### 3.2. Impact of Deformation on Microhardness

The results of measurements in the form of average values of microhardness and their percentage change for the parallel and perpendicular settings of the indenter are presented in Table 4. For measurements perpendicular and parallel to the direction of deformation, the tendency of microhardness change with deformation was similar. Microhardness decreased with the deformation. For this reason, the graphs are limited to showing the results of parallel direction measurements (Figure 12).

Analysing the results of microhardness measurements, it can be noticed that the decrease in microhardness was observed already with a small deformation *ε* = 2%. During the test, the microhardness meter indenter was positioned parallelly and perpendicularly to the direction of the tensile force vector. The differences between the microhardness values obtained in both cases were insignificant. The change in microhardness is the greatest in the strain range 0–2% and becomes smaller as the strain increases.

Similar to the wear tests, microhardness tests were carried out on samples maintained in deformation for 24 h. The microhardness of the samples was measured before deformation, immediately after deformation, and after maintaining deformation for 24 h.

The results are presented in the form of a table (Table 5) and a graph (Figure 13) presenting microhardness values and percentage changes in relation to the value obtained for an undeformed sample. Microhardness of PE-HD after 24 h increases by about 21% in relation to the value obtained in the measurement directly after deformation. The microhardness of PE-HD is lower by 23% in comparison to an undeformed sample.

### 3.3. Effect of Deformation on the Surface Free Energy of PE-HD

The results of surface free energy measurements of polyethylene PE-HD showed a slight influence of deformation on the change of its value. The results for which the mean value obtained for a given deformation was higher by 10% or more than the mean value obtained for a pair containing an undeformed polymer are highlighted in orange in Table 6. Such a change in the value of surface free energy was accepted as significant because of the expanded uncertainty range, which was ± 5% of the mean value.

For PE-HD after tension to 20% and to 50% strain, the surface free energy increased by 8% and 6%, respectively (compared to the value obtained for the undeformed sample). Maintaining the deformation caused an increase in the value of free surface energy by about 10%, which in the end constituted about 16% more than for the undeformed sample (Figure 14).

### 3.4. Microscopic Observations of the Sample Surface After Friction

Figure 15 shows photographs taken after the friction process, showing undeformed and deformed PE-HD samples. The obtained images are presented by juxtaposing photographs of undeformed and deformed samples. In the case of undeformed PE-HD polyethylene, the size of wear products did not exceed 50 µm (Figure 15a). The surface after friction shows grooves/furrows directed parallel to the direction of the interaction, which may indicate the dominance of abrasive wear (Figure 15b,c).

In the terms of PE-HD deformed to *ε* = 20%, the size of wear products was much greater as compared to undeformed polymer (Figure 15d). They were “pressed/dented” into the surface layer of the material. Observing the sliding surface of a deformed polymer, it seems as if it is “rubbed/spread” on the surface during interaction with a steel roll (Figure 15e,f).

## 4. Discussion

### 4.1. Correlation between Wear, Microhardness, and Surface Free Energy Test Results

An attempt was made to explain how the deformation affects the wear of PE-HD. Thus, the results of tribological studies, microhardness, and surface free energy (Table 7) were compared. Additionally, microscopic images were commented on and the issue of microstructural defects was raised. Analyses of the results of wear and microhardness tests revealed a correlation between values. As known from the literature, hardness plays a significant role in wear resistance [18]. It is a common rule that harder materials possess higher wear resistance. Furthermore, it is noteworthy that proportionality coefficient in the relation between hardness and wear is 3.2 larger for polymers than for metals.

When analysing the changes in the wear coefficient after maintaining the polymer in the deformation for a certain period of time, a correlation between the microhardness and the wear coefficient can be observed (Table 8). Namely, microhardness increases when the polymer is maintained in a deformed state. At the same time, the wear coefficient is increasing.

Adhesive wear and friction transfer are significant phenomena during wear of polymers [19]. Due to adhesion small particles are transferred from polymer surface to metal surface. In order to assess strength of adhesive contacts, surface free energy was quantified. It was observed the value of free surface energy increases after maintaining the polymer in a deformed state. The wear coefficient also increases with time. It appears that the increase of surface free energy in PE-HD causes an increase in the share of adhesive wear to such an extent that it eliminates the effect associated with the increase in microhardness. Therefore, the question of the link between microhardness and wear is not clear.

The research presented in the literature has shown that, in terms of polyethylene, deformation under tension to a strain value ε ≥ 500% resulted in an increase in the degree of crystallinity and a significant reduction in wear [6]. Moreover, increasing wear with reducing degree of crystallinity for various polyethylenes was proven in the research [20]. The increase in degree of crystallinity ought to be associated with an increase in the cohesion forces of the polymeric material. Therefore, reduced cohesion makes it easier to pull out pieces of material during interaction with a steel roll.

In the current research, the degree of crystallinity was not determined but the microhardness was measured. The decrease in microhardness is associated with reduction in the degree of crystallinity for semicrystalline polymers [21]. Presumably, deformation under tension in the initial phase reduces the degree of crystallinity. As a result, wear is increased. When deformation reaches vast values (several hundred percent), structure becomes highly oriented, the degree of crystallinity increases, and wear is reduced.

Changes in microhardness have a higher percentage than changes in surface energy. This may lead to the conclusion that the modification of cohesion forces has a greater impact on the increase of wear. It ought to be noted, however, that the adhesive wear is also involved in the process of polymer-metal wear. Therefore, small changes in surface energy can lead to significant changes in wear.

### 4.2. Defects due to Tension vs Wear

When stretching at room temperature to a strain of several dozen percent, various phenomena and processes occur in semi-crystalline polymers. For small values of deformation, local slippages within the lamellas and between the lamellas were observed, and after exceeding the yield point, the set of local slippages transforms into a widespread crystallographic slippage and inter-lamellar shear [22]. The separation of crystalline areas from each other causes the formation of empty spaces in the amorphous zone of so-called cavitation [23]. The cavities are stable (they do not become closed) immediately after reaching the yield point, and they have an elliptical shape (for PE-HD stretched to deformation 100% the minor semi-axis is 22 ÷ 40 nm) [24]. The cavities are arranged next to each other, separated by fibrils, and create stress cracks. They are the precursors of fissures. Stress cracks are mainly associated with amorphous polymers, but they also appear in semi-crystalline polymers (e.g., in PE-HD) [25]. Furthermore, researchers point out that, after PE-HD injection, the top-layer had a different structure, and therefore, in the top layer, the cavities appeared already at the deformation ε = 0.15% [26].

Tests conducted earlier [27] showed that, at 5% strain, an increase in wear was observed for tension and not for compression. Therefore, it can be presumed that increased wear is associated with phenomena characteristic of tensile stress. Cavities and stress cracks are only observed in tension. Thus, it is likely that these defects appearing in the polymer contribute to increased wear. This is confirmed by the studies presented in the literature, in which the influence of cavities on the wear of tensile fibre was indicated [10]. It appears to be highly possible that defects of this kind support the separation of successive pieces of material during wear. In addition, wear takes place in the near-surface layer, where defects of the polymer material may occur even at slight deformations [26].

### 4.3. Analysis of Microscopic Images

Microscopic images of the sliding surface of the polymer show the differences between the variant where the polymer was undeformed and the variant where it was stretched. Firstly, on the surfaces of PE-HD samples deformed by tension, “creases, crinkles” or “worn-out flakes” of the polymer were observed. These observations seem to indicate that the material under investigation is plasticized. Secondly, on the sliding surface of an undeformed PE-HD sample, grooves and furrows were observed which were not so visible on the surface of the deformed polymer. It can be concluded that the tension of the sample caused a decrease in the share of abrasion in the process of wear. This is confirmed by a photo of the wear products. For deformed PE-HD, the wear products are pressed into the polymer and are larger in size than for undeformed polymer.

### 4.4. Further Research

The tests were carried out for unmodified polymers. However, polymer composite materials are also used in technical applications. Therefore, another research ought to be conducted to determine the influence of deformation for this type of material.

A complex deformation state occurs in the sliding elements subjected to deformation during assembly of machines. In the research presented in this work, measurements were carried out for only uniaxially stretched and compressed materials. Therefore, it is worth carrying out research in the future to clarify whether the complex state of deformation in polymeric materials has an effect on friction and wear.

The wear tests carried out in the described research were conducted using the roller-plate system in which pressure was concentrated on a small area. In terms of lip seals, the pressure is distributed over the entire perimeter, where the seal contacts the shaft. In the further research, the wear test ought to be conducted with evenly distributed pressure.

The research on the structure of the polymer material ought to be conducted. The research ought to concern the determination of the change in the degree of polymer crystallinity or identification of voids and crazes appearing during deformation. The testing of polymers in a deformed state requires maintaining deformation of the samples. This involves the design and implementation of special additional equipment adapted to the measuring device. Due to these difficulties, such studies have not been carried out yet.

## 5. Conclusions

The results of tests carried out within the framework of this study have shown that tension applied to PE-HD contributes to the change of friction coefficient and wear coefficient during interaction with steel. The tension of PE-HD leads to increased wear and reduced microhardness of the polymer. In addition, such phenomenon pose higher threat as these changes will occur at 2% strain, which is usually considered acceptable in the design recommendations.

Deformation by tension significantly affects, among other things, the wear of a polymer in the interacting pair: a steel roll–polymer plate. The wear coefficient in relation to the value obtained for the undeformed polymer was even 5.5 times larger (ε = 20%).

Changes in polymer properties occur already at deformation resulting from tension up to 2% strain. In this situation, microhardness decreased by 17% and wear increased by 51%. This is important information as the deformation range up to 2% strain is considered safe and acceptable when designing various elements.

The direction of interaction in relation to the tension force vector is important. Greater changes concern the sliding interaction in the parallel direction. The tensile wear of PE-HD was significantly higher when applied parallelly to the direction of the tensile force than when applied perpendicularly.

Maintaining PE-HD in a deformed state with applied tension for a certain period of time results in a different material behaviour than in measurements carried out immediately after deformation. In the terms of PE-HD, after a time, the wear was higher than in the case of measurements carried out directly after deformation. This is important, for instance, for seals that are deformed during assembly and function as they are during their lifetime. The described effect does not diminish with time; the seal can be characterized by increased wear even after a considerable period of time after assembly.

In terms of polymer properties, the deformation is unfavourable. It causes a significant increase in wear and a decrease in microhardness. The deformation of a polymer material does not appear to be a way to improve its properties. It ought to be treated as a threat that must be taken into account when designing sliding elements that are strained during assembly.

After deformation due to tension, the changes include both friction components describing the metal-polymer interaction, i.e., the mechanical and adhesive component. Any attempt to explain the effect of tension on friction and wear ought to refer to both of these components.

## Figures and Tables

**Figure 1 polymers-11-01429-f001:**
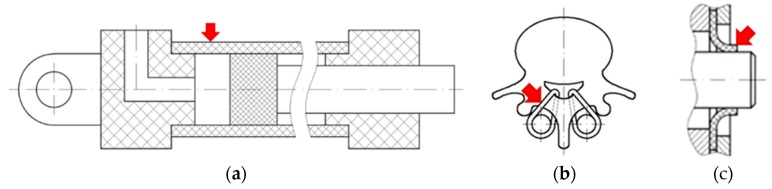
Deformed parts (indicated by red arrows) made of polymeric materials interacting frictionally: (**a**) hydraulic cylinder tube; (**b**) spine stabilising wires; and (**c**) shaft lip seal.

**Figure 2 polymers-11-01429-f002:**
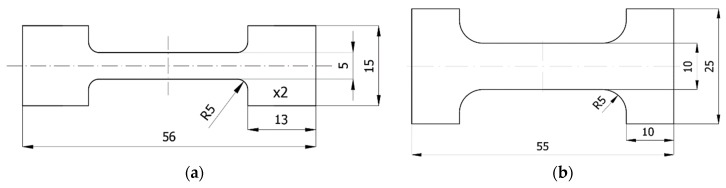
Sample dimensions used for measurements during tension: (**a**) interaction parallel to the direction of the tensile force vector and (**b**) interaction perpendicular to the direction of the tensile force vector.

**Figure 3 polymers-11-01429-f003:**
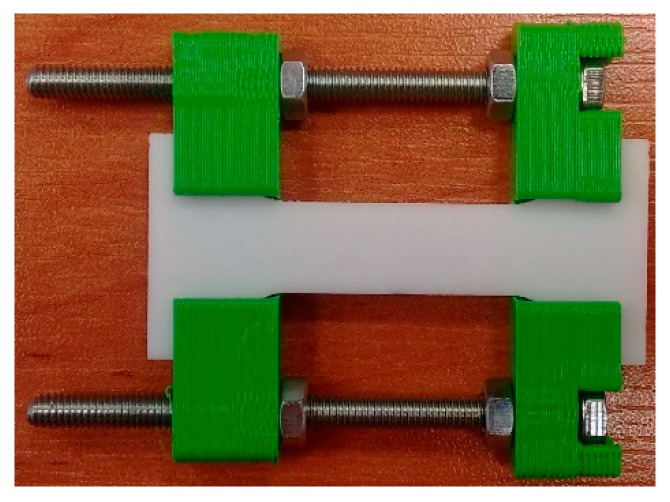
Sample grip for stretching samples and maintaining deformation during measurements: The callipers of the holder were made using a 3-D printer.

**Figure 4 polymers-11-01429-f004:**
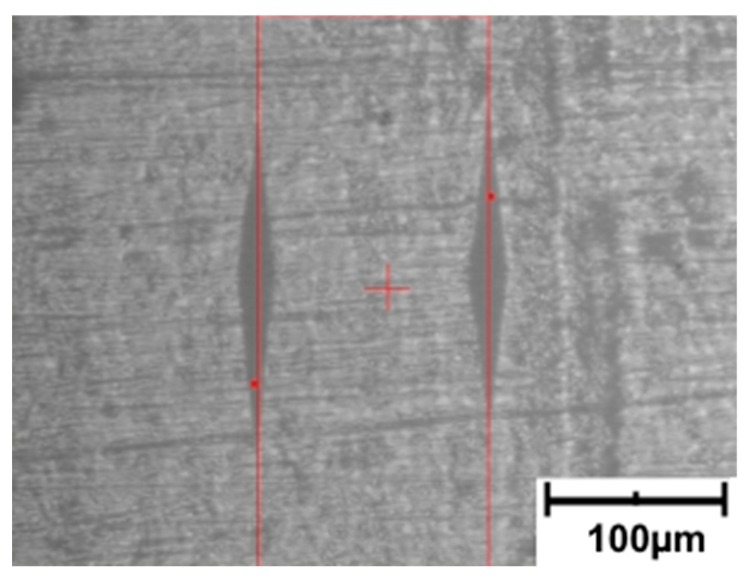
Microscopic image used to determine the distance between the imprints.

**Figure 5 polymers-11-01429-f005:**
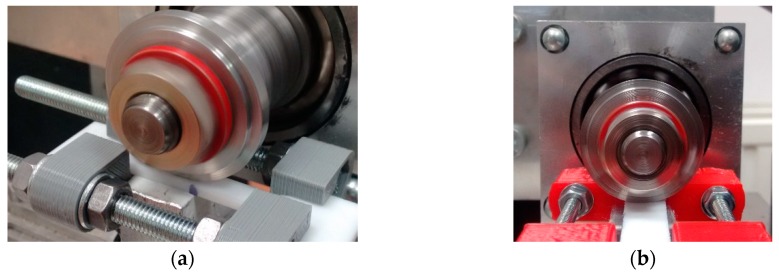
The combination of stretched polymer–steel roller when measuring wear in a direction: (**a**) parallel and (**b**) perpendicular to the direction of the tensile force vector.

**Figure 6 polymers-11-01429-f006:**
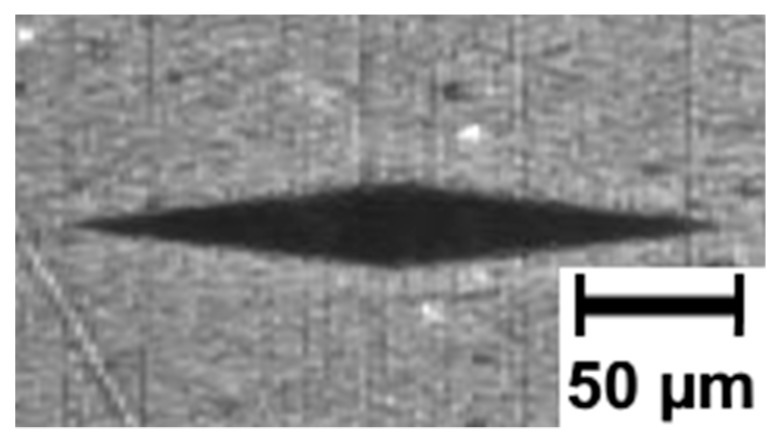
Imprint obtained from the Knoop microhardness test.

**Figure 7 polymers-11-01429-f007:**
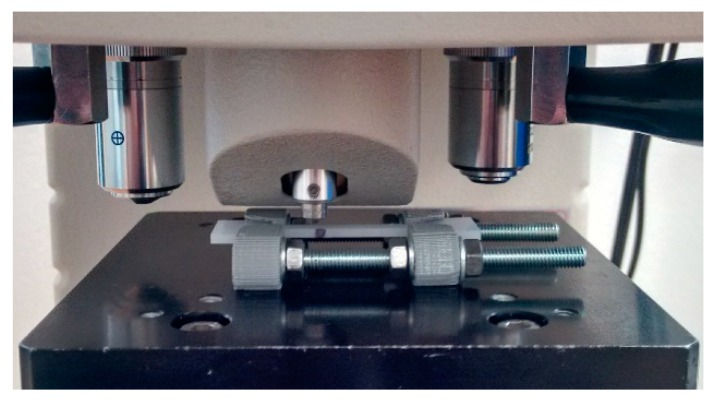
Measurement of microhardness of a deformed polymer subjected to tension.

**Figure 8 polymers-11-01429-f008:**
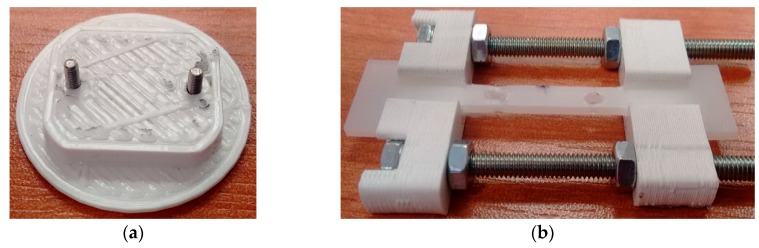
(**a**) Grip adjusted to SEM slot; (**b**) sample with holes made after applying tension; (**c**) stretched sample attached to the grip; and (**d**) deformed polymer placed in the microscope slot.

**Figure 9 polymers-11-01429-f009:**
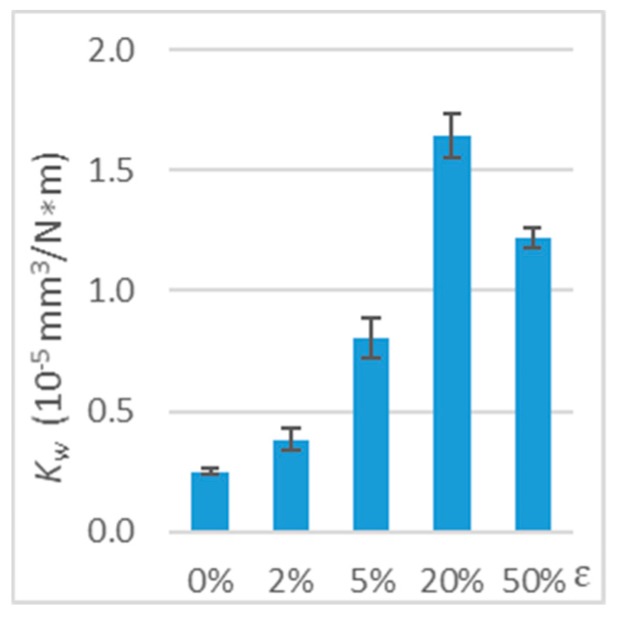
Wear coefficient (*K*_w_) for deformed polymer (working with steel, tension, parallel direction, *v* = 0.33 m/s, and *T_0_* = 23 °C).

**Figure 10 polymers-11-01429-f010:**
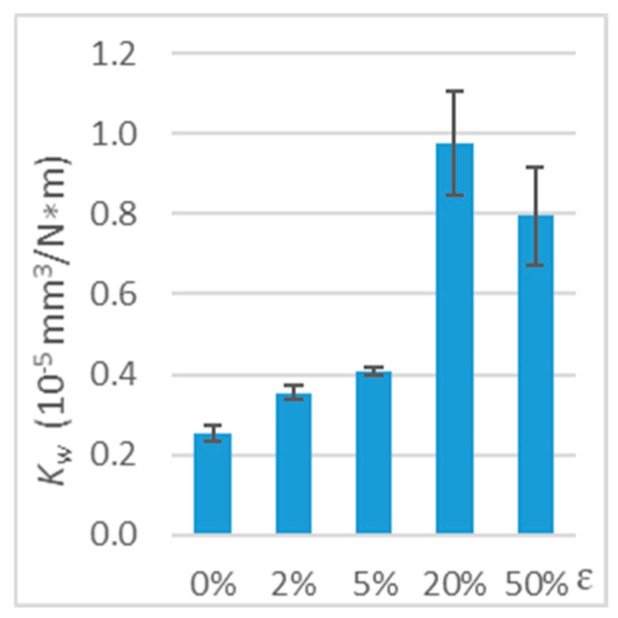
Wear coefficient (*K*_w_) of deformed polymer (working with steel, tension, perpendicular direction, *v* = 0.33 m/s, and *T*_0_ = 23 °C).

**Figure 11 polymers-11-01429-f011:**
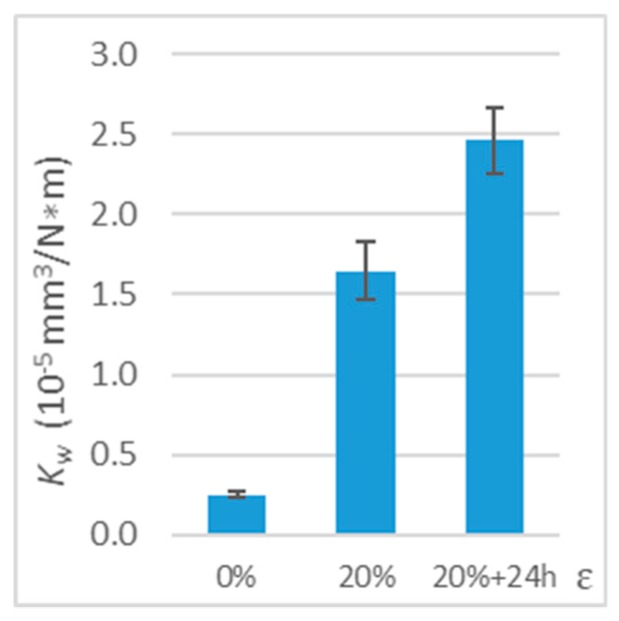
Wear coefficient (*K*_w_) for deformed polymer (working with steel, tension, parallel direction, 24h, *v* = 0.33 m/s, and *T*_0_ = 23 °C).

**Figure 12 polymers-11-01429-f012:**
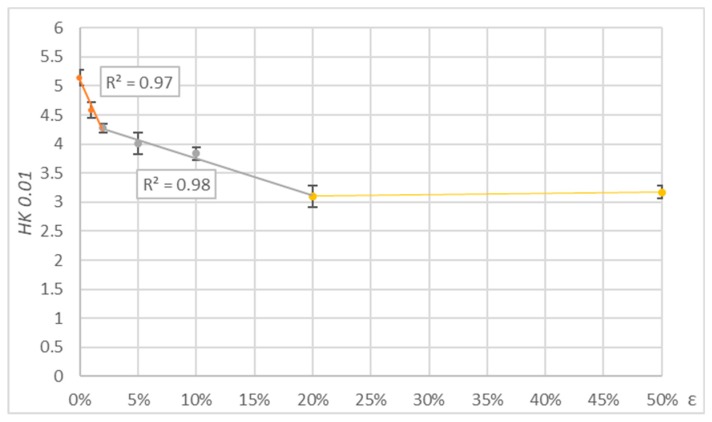
Results of microhardness tests for PE-HD deformed by tension.

**Figure 13 polymers-11-01429-f013:**
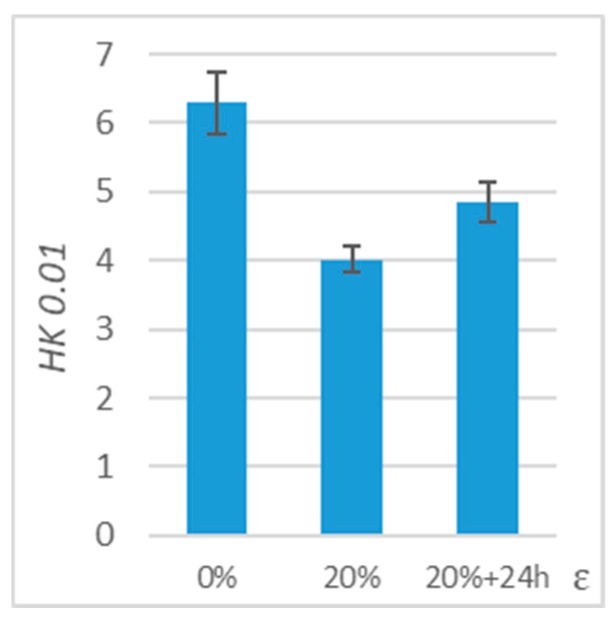
Results of microhardness tests for PE-HD maintained in deformation for 24 h.

**Figure 14 polymers-11-01429-f014:**
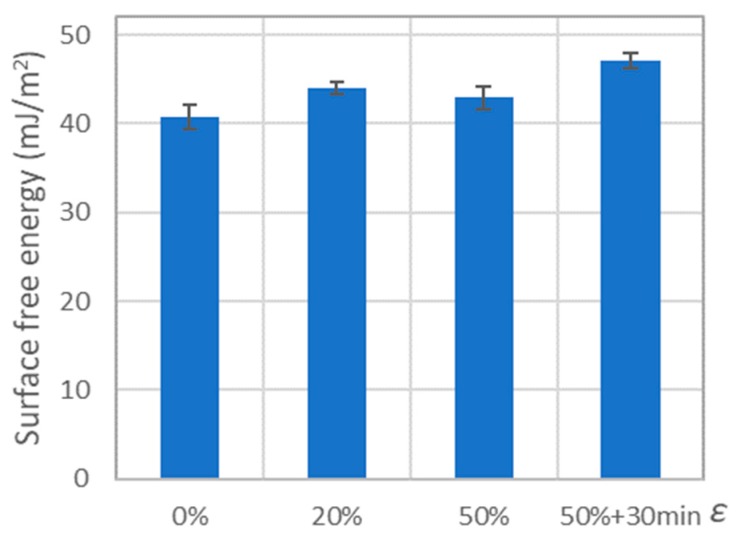
Results of surface free energy tests for PE-HD.

**Figure 15 polymers-11-01429-f015:**
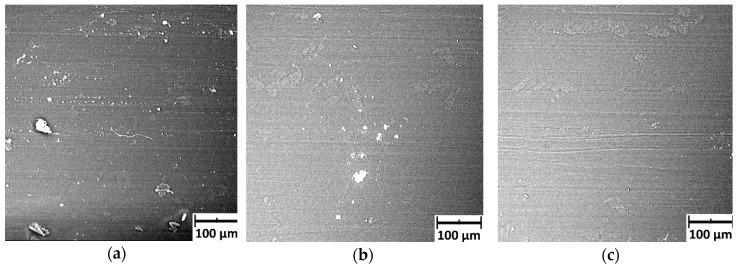
Selected SEM microscopic images of PE-HD surface after frictional interaction with a steel roll: (**a**–**c**) undeformed polymer and (**d**–**f**) polymer deformed to ε = 20%.

**Table 1 polymers-11-01429-t001:** Test results on the wear of deformed polymer (working with steel, tension, parallel direction, *v* = 0.33 m/s, and *T*_0_ = 23 °C.

*ε*	0%	2%	5%	20%	50%
*V* (mm^3^)	0.25 ± 0.02	0.37 ± 0.09	0.79 ± 0.16	1.61 ± 0.18	1.20 ± 0.08
*K*_w_ ($\frac{10^{-5}\;mm^{3}}{N*m}$)	0.25 ± 0.02	0.38 ± 0.09	0.80 ± 0.16	1.64 ± 0.18	1.22 ± 0.08
*ΔK* _w_	-	+51%	+220%	+554%	+386%

**Table 2 polymers-11-01429-t002:** Test results on the wear of deformed polymer (working with steel, tension, perpendicular direction, *v* = 0.33 m/s, and *T_0_* = 23 °C).

*ε*	0%	2%	5%	20%	50%
*V* (mm^3^)	0.25 ±0.02	0.35 ±0.02	0.40 ±0.01	0.96 ±0.13	0.78 ±0.12
*K*_w_ ($\frac{10^{-5}\;mm^{3}}{N*m}$)	0.25 ±0.02	0.35 ±0.02	0.41 0.01	0.98 ±0.13	0.79 ±0.12
*ΔK* _w_	-	+41%	+62%	+289%	+216%

**Table 3 polymers-11-01429-t003:** Test results on the wear of deformed polymer (working with steel, tension, parallel direction, deformation maintained for 24 h, *v* = 0.33 m/s, and *T*_0_ = 23 °C).

*ε*	0%	20%	20% + 24 h
*V* (mm^3^)	0.25 ± 0.02	1.61 ± 0.18	2.42 ± 0.20
*K*_w_ ($\frac{10^{-5}\;mm^{3}}{N\;m}$)	0.25 ± 0.02	1.64 ± 0.18	2.46 ± 0.21
*ΔK* _w_	-	+554%	+879%

**Table 4 polymers-11-01429-t004:** Results of microhardness tests for polymers deformed by tension.

Direction of Indenter		ε = 0%	ε = 2%	ε = 5%	ε = 20%	ε = 50%
Parallel	*HK 0.01*	5.14 ± 0.12	4.27 ± 0.15	4.01 ± 0.17	3.31 ± 0.16	3.15 ± 0.22
	*ΔHK 0.01*	-	−17%	−22%	−35%	−38%
Perpendicular	*HK 0.01*	4.72 ± 0.18	4.05 ± 0.14	3.69 ± 0.09	3.09 ± 0.18	3.10 ± 0.21
	*ΔHK 0.01*	-	−14%	−22%	−35%	−34%

**Table 5 polymers-11-01429-t005:** Results of microhardness tests for polymers maintained in deformation for 24 h.

*ε*	0%	20%	20% + 24 h
*HK 0.01*	6.3 ± 0.45	4.02 ± 0.18	4.85 ± 0.30
*ΔHK 0.01*	-	−36%	−23%

**Table 6 polymers-11-01429-t006:** Results of surface free energy tests for deformed polymers.

*ε*	0%	20%	50%	50% + 30 min
*SFE* (mJ/m^2^)	40.7 ± 1.4	44.0 ± 0.6	42.9 ± 1.2	47.1 ± 0.8
*ΔSFE*	-	+8%	+6%	+16%

**Table 7 polymers-11-01429-t007:** Summary of test results of wear (*K*_w_), surface free energy (*SFE),* and microhardness (*HK 0.01*) for tensile deformed PE-HD.

ε	0%	2%	5%	20%	50%
*K*_w_ ($\frac{10^{-5}\;mm^{3}}{N\;m}$)	0.25 ± 0.02	0.38 ± 0.09	0.80 ± 0.16	1.64 ± 0.18	1.22 ± 0.08
*ΔK* _w_	-	+51%	+220%	+554%	+386%
*HK 0.01*	5.14 ± 0.12	4.27 ± 0.15	4.01 ± 0.17	3.31 ± 0.16	3.15 ± 0.22
*ΔHK 0.01*	-	−17%	−22%	−35%	−38%
*SFE* (mJ/m^2^)	40.7 ± 1.4	X	X	44.0 ± 0.6	42.9 ± 1.2
*ΔSFE*	-	X	X	+8%	+6%

**Table 8 polymers-11-01429-t008:** Summary of test results on wear (*K*_w_), surface free energy (*SFE),* and microhardness (*HK 0.01*) for PE-HD maintained in deformation for a time.

ε	0%	20%	20% + 24 h
*K*_w_ ($\frac{10^{-5}\;{mm}^{3}}{N\;m}$)	0.25 ± 0.02	1.64 ± 0.18	2.46 ± 0.21
*ΔK* _w_	-	+554%	+879%
*HK 0.01*	6.3 ± 0.45	4.02 ± 0.18	4.85 ± 0.30
*ΔHK 0.01*	-	−36%	−23%
*ε*	0%	50%	50% + 30 min
*SFE* (mJ/m^2^)	40.7 ± 1.4	42.9 ± 1.2	47.1 ± 0.8
*ΔSFE*	-	+6%	+16%

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
