# Peer review of "The Influence of Deformation under Tension on Some Mechanical and Tribological Properties of High-Density Polyethylene"

_polymers, 2019, doi:10.3390/polym11091429_

Round 1

Reviewer 1 Report

In this article, the authors investigated the effects of deformation of PE-HD on the wear. The authors showed that the wear of deformed PE-HD was higher than the wear of undeformed sample. Even after PE-HD was maintained as deformation for 24 hours, the effect on the wear was still significant. In addition, microscopic observations of PE-HD after the friction, measurements of microhardness and free surface energy were investigated to explain the results of the wear. The authors suggested that the deformation of a polymer material should be treated as a hazard instead of as a way to improve the property of a polymer material. Although the authors did provide supportive results and addressed the potential mechanisms, the authors need to address the following concerns:

There was too much information in the introduction, some literature reviews should be removed to the discussion. The introduction should make readers easy to understand the rationale of the study. It is difficult to understand the figure 1. What do the red arrows mean? The authors explained the reasons of conducting each experiment in the section of materials and methods. Perhaps the reasons should be removed to the results or discussion (such as line 235-242), and simply describe how to conduct each experiment in materials and methods. In discussion, the authors mostly explained the potential mechanism of the results they observed without providing the supports of other research. More comparisons of this study and the literature are advised. What are the weaknesses of this study? The authors should acknowledge the weaknesses in the discussion. The authors should correct some wordings that are too conversational. For example, line 59 “thanks to which strain and stress….”In addition, “g”was also used a lot when sometimes was not necessary. Line 122, 3D should be written as three-dimensions.

Reviewer 2 Report

The paper is interesting and well written, thus in my opinion it deserves publication. I only ask the authors to give more information about the material, that is not just the type of polymer but also the commercial name, if possible. Lastly, as far as I know, wear in polymers is dependent on the heat generation due to sliding contact. It would be interesting to check whether there is correlation between temperature and prestrain, maybe as a future development

Round 2

Reviewer 1 Report

The authors have addressed all the concerns, and the manuscript has been edited to be easy to understand.